# Reliability of Standardised High-Intensity Static Stretching on the Hamstrings over Multiple Visits

**DOI:** 10.3390/muscles4030033

**Published:** 2025-08-15

**Authors:** Joseph Bryant, Darren J. Cooper, Derek M. Peters, Matthew D. Cook

**Affiliations:** 1School of Sport and Exercise Science, University of Worcester, Henwick Grove, Worcester WR2 6AJ, UK; bryj1_21@uni.worc.ac.uk (J.B.); d.cooper@worc.ac.uk (D.J.C.); 2Independent Researcher, Worcester WR, UK; dmpeters99@gmail.com

**Keywords:** flexibility, range of motion, variability, intensity

## Abstract

Static stretching (SS) is commonly used in athletic programs, and the intensity of SS has recently been examined for its effects on range of motion (ROM), strength and passive stiffness. However, the reliability of high-intensity SS across multiple testing sessions has not been investigated. The purpose of this investigation was to examine the reliability of high-intensity SS of the hamstrings across five laboratory visits on ROM, strength, power and passive stiffness. Thirteen physically active males (age: 26 ± 4 years, height: 180 ± 8 cm, body mass: 81 ± 10 Kg) underwent five repeated measures of laboratory SS on an isokinetic dynamometer where point of discomfort (POD) was measured, followed by a 30 s stretch at 120% POD. Across the visits, the pooled intraclass correlation coefficient was good for knee extension ROM (0.82), knee flexion strength (0.81) and passive stiffness (0.81). The ROM achieved to determine the POD before the SS was not different for the five visits (*p* = 0.370). These findings suggest high-intensity SS to 120% POD on an isokinetic dynamometer is reliable across multiple testing sessions. It is not clear if high-intensity static stretching is also reliable within applied scenarios and warrants further investigation.

## 1. Introduction

Static stretching (SS) is used to increase flexibility and range of motion (ROM) and decrease musculotendinous unit (MTU) stiffness [1]. Flexibility is also identified as a key component of health and should be included in exercise promotion [2]. However, there can be post-stretch reductions in force and power output, for example, Kay and Blazevich [3] identified in a systematic review a curvilinear relationship between static stretch duration and a reduction in strength, power and speed-dependent tasks with decrements mostly observed following stretch durations of more than 60 s. However, this review did not consider if the intensity of SS was an important factor in determining magnitude of reductions. Because of these effects, The European College of Sports Science, the American College of Sports Medicine and the Canadian Society for Exercise Physiology do not recommend the use of SS as part of a warm-up routine and instead promote dynamic stretching for warm-up routines [2,4,5].

Altering the duration and volume of SS is objective, quantifiable and simple to amend for athletes and coaches. However, intensity is subjective and has received inconsistent definitions in the literature. For example, Jacobs and Sciascia [6] defined it as “The magnitude of force or torque applied to the joint during a stretching exercise.” However, Freitas et al. [7] defined it as “The degree of muscle-tendon lengthening induced by a change in joint range of motion”. Furthermore, examining intensity has also been approached inconsistently in the methods of different studies [8]. For example, some studies ask participants to identify “point of discomfort” [9], “point of pain” [10] or a verbal numerical scale [11]. This is likely to add to the variability in outcomes from these studies, as the terminology used when stretching the joint to the maximal range of motion is important for the final joint angle [12]. What is more, the findings from a recent systematic review of studies examining high-intensity static stretching identified inconsistencies in the results, and it is not clear whether higher -stretching for short durations leads to greater changes in ROM or strength and power reductions [8]. Furthermore, this review also identified that lots of studies use cross-over designs; therefore, the reliability of stretch sensation could influence these results, especially as temperature, muscle damage and menstrual hormones can affect ROM [8]. This review also recommends that static stretching studies report changes from a standardised non-stretching reference point so that outcomes between studies can be assessed [8].

The mechanisms for an increase in ROM following acute SS is attributed to a less stiff MTU [13]. When SS is undertaken chronically, this may influence injury risk [14] as high stiffness reduces the cushioning effect of soft tissue, resulting in greater stress [15]. Furthermore, changes in strength can also be caused by alterations in a less stiff MTU and lower motor unit activation. The repeatability of SS to influence strength over multiple visits is important for those prescribing SS.

It is likely SS will be used within numerous strength and conditioning regimes. The intensity of resistance exercise sessions can be quantified repeatedly with a high level of accuracy [16]; however, the intensity of SS is not as easily quantifiable between sessions. Therefore, the variability of achieving a high-intensity stretch across multiple visits is unknown. Reliably stretching to achieve a high-intensity stretch may also have further important considerations because higher-intensity stretching has been demonstrated to lead to greater flexibility [17,18]. Flexibility assessment of the hip flexors is inconsistent [19]. The passive assisted assessment of flexibility has previously been shown to be reliable [20]; however this was not conducted on an isokinetic dynamometer. Therefore, multiple visits at the same time of day can identify if this method is reliable.

As multiple methods are used to describe high-intensity SS, it is possible that intensity in studies is inconsistent and results in a stretch above or below the target intensity. Furthermore, target intensity is often subjectively identified by participants in cross-over designs separated by a few days. As such, it is not currently known if these methods are reliable across multiple sessions. Therefore, the aim of this study was to investigate the reliability of high-intensity stretching across five repeat visits and comparing ROM, strength and power to a high-intensity stretch that was 120% point of discomfort (POD) (i.e., 120% past the point of the beginning of the stretching sensation). It was hypothesised that high-intensity SS would cause a decrease in strength and passive stiffness; however, over multiple visits, the point at which a high-intensity stretch is identified is not reliable.

## 2. Results

### 2.1. Reliability

The reliability and confidence intervals for knee extension ROM, knee flexion peak MVC force and passive stiffness are illustrated in Table 1.

### 2.2. ROM, Stretching Perception and Passive Stiffness

The ROM achieved to determine the POD before the SS was not different for the five visits (F_(2.260, 27.115)_ = 1.095, *p* = 0.370) (Figure 1A). As a result, there was also no difference in the degrees of high-intensity stretching across the five visits (F_(2.151, 25.817)_ = 1.2387, *p* = 0.295) (Figure 1B). However, the ROM achieved following the SS across the visits was different (F_(2.464, 29.564)_ = 3.685, *p* = 0.029, ηp2 = 0.235). The ROM on visit one was different to visit four (*p* = 0.008), and five (*p* = 0.024). Visit three was also different to visits four (*p* = 0.019) and five (*p* = 0.018) (Figure 1A). Participants’ perception of the stretching intensity on the visual analogue scale was not different across the visits (F_(2.311, 27.734)_ = 0.805, *p* = 0.0474) (V1: 55 ± 22, V2: 49 ± 25, V3: 57 ± 21, V4 52 ± 21, V5: 56 ± 23 mm).

The Δ change in ROM pre-to-post-static stretch for each visit was different (F_(4, 48)_ = 2.679, *p* = 0.043, ηp2 = 0.183) (V1: Δ5.5 ± 5.3, V2: Δ7.2 ± 7.4, V3: Δ5.8 ± 8.7, V4: Δ10.2 ± 4.9, V5:10.8 ± 5.1°), with the change following visits four (*p* = 0.018) and five (*p* = 0.021) being different to visit one.

The peak torque achieved during the stretch was different over time (F_(4, 48)_ = 3.675, *p* = 0.011, ηp2 = 0.234), with the torque achieved in visit two being lower than visits three (*p* = 0.017), four (*p* = 0.022) and five (*p* = 0.006) (Figure 2A). Following the stretching, the gravity-corrected passive stiffness achieved was not different across the visits (F_(4, 48)_ = 0.492, *p* = 0.742) (Figure 2B).

### 2.3. Strength

There was no difference in the MVC produced before the high-intensity static stretch across the visits (F_(2.352, 28.331)_ = 1.494, *p* = 0.241). Similarly, the MVC produced following the 30 s high-intensity stretch was also not different across the visits (F_(4, 48)_ = 0.761, *p* = 0.556) (Figure 3). The Δ change in strength for pre-to-post-static stretch from each visit was different (F_(4, 48)_ = 3.227, *p* = 0.020, ηp2 = 0.212) (V1: Δ5.2 ± 17.8, V2: Δ-1.2 ± 11.8, V3: Δ-2.6 ± 9.1, V4: Δ-12.0 ± 12.1, V5-3.5 ± 9.9 Kg), with changes following visits one and four (*p* = 0.003), two and four (*p* = 0.025), three and four (*p* = 0.049) and four and five (*p* = 0.026).

### 2.4. Drop-Jump Peak Force

The peak force from the 20 cm drop single-leg vertical jump before stretching was not different (F_(4, 48)_ = 0.206, *p* = 0.934) (Figure 4A), and neither was the peak force achieved on the fourth maximal single-leg jump (F_(4, 48)_ = 1.536, *p* = 0.207) (Figure 4B). Following the high-intensity SS, similarly, the peak force during the drop jump was not different (F_(4, 48)_ = 1.389, *p* = 0.252) (Figure 4A), and neither was the peak force achieved on the fourth jump (F_(4, 48)_ = 0869, *p* = 489) (Figure 4B).

## 3. Discussion

The main finding of this study is that high-intensity stretching at 120% over five visits 72 h apart is reliable in physically active young males. However, the increase in the ROM as measured by the POD by SS increased, with the highest post-stretching ROM achieved following visit five. To the authors’ knowledge, this is the first study to examine the reliability of high-intensity SS across multiple visits using an isokinetic dynamometer to perform the stretching.

The increase in the post-stretching POD across time potentially could indicate the beginning of a training response in relation to multiple stretching visits. However, SS was only performed for 30 s during each visit, and therefore the participants performed a total (i.e., duration x umber of sessions) of 150 s high-intensity SSs. Thomas et al. [21] identified that when stretching between 5 and 10 min, or more than 10 min per week, there is an increase in the ROM. However, the analysis by Thomas et al. [21] did not consider the influence of intensity on the required volume of stretching to elicit an increase in ROM. Furthermore, the studies included in their analysis used training periods in the range of 4–16 weeks. On the whole, the present study uses values lower than the weekly five-minute volume; therefore, it is unlikely that the repeated stretching in the five visits was a large enough stimulus to mechanically increase the ROM. Future research could investigate the outcome of repeated high-intensity stretching for longer than five minutes per week and if more intense SS brings about training adaptations quicker than less intense SS, but of the same duration.

The stretching sensation presented within the present study is comparable to previous studies. For example, the pain reported by participants on the visual analogue scale was ~50 mm for all the visits. Previous studies have reported stretching the quadriceps at 120% ROM to be 59.5 mm [11] and ~60 mm [9]. Therefore, stretching in the present study indicates that pain during SS is high. Furthermore, as pain reported on the VAS was unchanged across the visits, it confirms that participants reliably identified their POD and a high-intensity stretch. From an applied perspective, however, it should be recognised that, due to the subjective nature of pain, requiring participants with a high pain threshold to stretch to their POD could be placing greater stress on the tissues than for an individual with a lower threshold.

The mechanisms for a force loss following SS are multifactorial, but include reduced tendon stiffness resulting in the MTU functioning at a shorter and weaker part of the length–tension relationship [22]. It was outside the scope of this experiment to investigate the underpinning mechanisms; however, this experiment does show that the MVC produced following the 30 s stretch was not different across the visits (Figure 3). Interestingly, there was also no change in the drop jump and fourth hop jump following stretching (Figure 4A,B). Previous observations have demonstrated decreases in power following static stretching [23]. It is not clear if this indicates that the 30 s 120% POD stretch in the present study is not a sufficient duration to have a negative impact on power generation or error of the measurement. Therefore, future observations need to clarify if intense static stretches of short durations impacts power generation.

Increases in ROM following SS are attributed to decreases in passive stiffness [24]; however, this study demonstrated that the gravity-corrected passive stiffness remained unchanged following static stretches (Figure 2B). This suggests that the increase in ROM following a high-intensity static stretch may be due to an increase in stretch tolerance [25,26]. The lack of change in passive stiffness following a static stretch may be due to the duration of the stretch (i.e., 30-s), and it is possible that shorter-duration stretches (<60 s) may be insufficient to reduce passive stiffness [27,28]. Matsuo et al. [27] compared different durations of hamstring static stretches on passive stiffness and found that passive stiffness did not decrease after 20 and 60 s of stretching and only decreased after 180 s and 300 s of stretching. Furthermore, Stafilidis et al. [28] examined 15 s and 60 s of SSs of the quadriceps and showed that neither stretching durations resulted in decreases in passive stiffness. Lastly, Santos et al. [29] examined three sets of 60 s high-intensity SSs in the hamstring group on ROM and passive stiffness. ROM increased (pre: 130.5 ± 15.9, post: 133.4 ± 14.9°); however, passive stiffness remained unchanged (pre: 0.52 ± 0.09, post: 0.57 ± 0.13 Nm·°^−1^), further suggesting that ROM increases are due to increases in stretch tolerance.

### Limitations

The results of this study only apply to physically active young men. Therefore, it is not known if reliability is affected in clinical populations or females. The use of females when examining reliability over multiple visits could have added unknown variability results, especially as it is unclear if menstrual-cycle hormone fluctuations alter joint laxity [30,31]. Elliot-Sale et al. [32] identified the importance of including women in sport and exercise science research to support sex-specific guidelines. However, as this study examined reliability, it was important to have a homogenous sample, and females have a longer hamstring length and greater flexibility than males [33], which could potentially influence the outcomes. Future research must therefore compare males and females and outcomes on reliability, and examine if menstrual-cycle hormones affect outcomes in females.

While the participants in the current study were moderately active, none of them were elite athletes, and studies have shown that elite athletes may respond differently to acute SS [22,34]. Therefore, the reliability of the 120% POD method should be tested using elite athletes from sports that require high degrees of ROM, like dancers or gymnasts. Another limitation is that this study only examined the effects on the hamstrings. Future research should be conducted on other muscle groups; however, for other muscles, such as the quadriceps, it may not be biomechanically possible (i.e., structural limits) to stretch to 120% of the POD ROM.

A further limitation to the present investigation is that pain tolerance can vary between individuals. As a result, the ROM achieved to the POD is affected by the participants’ stretching tolerance. In turn, the measurement of passive torque provides an objective measure, and it should be recognised that this was not consistent across time. For visits two and three, this result was lower (Figure 2A).

## 4. Materials and Methods

### 4.1. Study

Ethical approval was granted for this study by the University of Worcester’s College of Business, Psychology and Sport Ethics Panel (CBPS22230006-R), with procedures conducted in accordance with the Declaration of Helsinki.

### 4.2. Participants

This study recruited 13 males (age: 26 ± 4 years, height: 180 ± 8 cm, body mass: 81 ± 10 Kg, BMI: 25 ± 3). Participants were briefed on this study’s aims and protocol and provided written informed consent. Participants completed a healthy history questionnaire to screen for inclusion in this study, and participants were excluded if they had lower limb injuries in the six months prior to this study, knee instability, hyper flexibility or neurological health conditions. Participants abstained from dietary supplements that could influence their recovery rate or muscular performance (i.e., protein supplements, creatine, beta-alanine, HMB). None of the participants recruited met the exclusion criteria or were taking dietary supplements. Participants were all physically active and met or exceeded the prerequisite of performing 150 min of physical activity per week, as determined by healthy history screening. While some participants undertook sport-specific training, all the participants were categorized as performance tier 1 (recreationally active) according to the guidelines from McKay et al. [35].

### 4.3. Experimental Design

Participants completed a repeated measures design for five visits, with a minimum of 72 h and a maximum of 96 h separating visits. This timing allowed for the scheduling of all visits for participants within a two-week period, and all participants were able to complete this within the schedule. An air-conditioned laboratory was maintained at 20 °C. The primary investigator [JB] responsible for conducting the static stretching protocol was a trained researcher and conducted all the testing sessions with the participants.

During visit one, participants became familiar with the protocols and procedures. For the procedures, participants firstly undertook 5 min of cycling at 60 revolutions per minute (~60 W) (Monark Ergomedic 874E, Monark Sports and Medical, Vansbro, Sweden). Participants were prohibited from undertaking any stretching routines as part of their warm-up. The baseline measurements of hamstring ROM, passive stiffness and maximal isometric strength (MVC) were then taken on an isokinetic dynamometer (Humac Norm Isokinetic dynamometer CSMi). Participants then dismounted the dynamometer and undertook vertical jump tests on a floor-mounted force plate. A 10 min rest period followed after the participants performed a 30 s high-intensity static stretch on the isokinetic dynamometer. Immediately after the stretch, participants recorded on a 100 mm visual analogue scale their perception of the stretch pain. The performance measures, ROM, passive stiffness, MVC and vertical jumps were then repeated in the same order.

### 4.4. Static Stretching Protocol

Participants were seated on an isokinetic dynamometer in a hip-flexed position, which has been shown to allow for a sufficient stretch of the hamstrings [27] (Figure 5). To achieve this position, the angle between the backrest and seat of the isokinetic dynamometer was set to 60°, with hip and chest straps used to prevent additional movement. The lateral epicondyle of the knee was aligned with the axis of rotation of the lever arm of the isokinetic dynamometer. To aid the set up and to identify bony landmarks, the participants were required to wear shorts for testing visits. The seating position setup was recorded on the first visit and replicated for all subsequent visits. The participants’ dominant leg was then passively extended by the primary investigator to 120% greater than the angle achieved in the pre-intervention ROM test (see Section 4.5); this position was then held for 30 s, with the participants being instructed to relax. Immediately following the 30 s stretch, the participants were asked to mark on a 100 mm visual analogue scale how they perceived the stretch from “No pain at all” to “worst pain imaginable”. For data interpretation, the participant’s mark was measured from 0 to 100 mm, with the interpretation that the higher the score, the higher the participant’s rated intensity of the stretch.

### 4.5. Range of Motion and Passive Stiffness

Participants were seated on the isokinetic dynamometer in the same position as all other protocols. The investigator passively moved the participants’ dominant leg into the knee extension position, and the participants were then instructed to indicate the first moment they felt a stretching sensation. This angle was used as their POD ROM angle. The dynamometer then electronically moved the participants’ leg from the start position (i.e., 0 degrees/vertical) to the POD angle and back to 5 deg·s^−1^ to measure passive stiffness. Throughout both procedures, participants were instructed to relax and not make any voluntary contractions. Feedback was not provided regarding the angle achieved. The anatomical 0° value was set at the angle where the participant’s tibia was vertical.

### 4.6. Strength Assessment

In the same flexed-hip position on the isokinetic dynamometer, the dynamometer moved the participant’s leg up to 50% of the angle achieved in the ROM test (i.e., 50% of POD). The maximal isometric force of the hamstrings was measured by participants’ flexing their knee as hard as they could for six seconds. Participants received standardised instructions [36] and strong verbal encouragement.

### 4.7. Power

Two single-leg vertical jump tests were performed to measure the power of the stretched leg on a floor-mounted force plate (AMTI BP600900 force plate with MSA-6 amplifier, AMTI, Watertown, MA, USA). Firstly, participants performed a single vertical jump from a 20 centimetre box by stepping off the box, landing on one leg and then jumping as high as they could. Participants were required to keep their hands on their hips to prevent arm swinging. Participants then rested for 60 s before performing three small non-maximal hops before the fourth jump that showed maximal effort. Data were acquired at 1000 Hz using a Vicon Lock Lab a-to-d unit and Vicon Nexus software 2.16 (Vicon, Oxford, UK), before being exported into Microsoft Excel for analysis.

### 4.8. Statistical Analysis

Statistical analysis was conducted using SPSS 29.01.0 (IBM SPSS Statistics, Armonk, NY, USA: IBM Corp). Data normality was assessed using a Shapiro–Wilk test, and no data violated the assumption of normality. Variables were analysed for time (i.e., visits 1–5), with a one-way repeated measures ANOVA. Repeated measurements were checked for sphericity violations using Mauchly’s test and, if violated, the Greenhouse–Geisser correction was applied. Where differences occurred, partial-eta2 (ηp2) was reported and followed by Bonferroni-corrected pairwise post hoc comparisons. Delta changes (Δ) in the pre-to-post-stretches were calculated for each visit, for ROM and strength. Intraclass correlation coefficients (ICCs) were calculated for each visit and compared to the previous visit (i.e., 2–1, 3–2, 4–3, 5–4). The pooled ICC was also calculated. The ICC was interpreted with the following criteria: <0.5 poor reliability, 0.5–0.75 moderate reliability, 0.75–0.9 good reliability and >0.9 excellent reliability. An alpha level of 5% was set for statistical significance. Using a power of 80%, an alpha error of 0.05 and effect size of 0.8 using G*Power software 3.1.9.7 (Heinrich Heine University Düsseldorf, Düsseldorf, Germany) demonstrated that 15 participants would be required for this study, with an expected dropout rate of 2. Data visualisation and presentation were performed on Graph Pad Prism 10.3.1 (GraphPad Software LLC, Boston, MA, USA, Dotmatics).

## 5. Conclusions

Across five laboratory visits, the range of motion achieved for the hamstrings during a 30 s static stretch on an isokinetic dynamometer to the POD was not different across multiple sessions. In addition, there was no change in participants’ perception of stretching intensity. Furthermore, there were also no changes in gravity-corrected passive stiffness and strength following SS. There was also good pooled reliability for range of motion, knee flexion strength and passive stiffness across multiple testing visits.

These findings will be useful for other researchers undertaking stretching investigations using isokinetic dynamometers, as the protocol is reliable. Furthermore, participants can reliably identify their point of discomfort when moving passively through a range of motion. For practitioners, these findings will be difficult to apply unless using isokinetic dynamometers in a rehabilitation or testing scenario. However, in research scenarios, the results of this study indicate that high-intensity static stretching on dynamometers is reliable and can be used across multiple visits, assuming that pre-experimental visit controls are utilized.

## Figures and Tables

**Figure 1 muscles-04-00033-f001:**
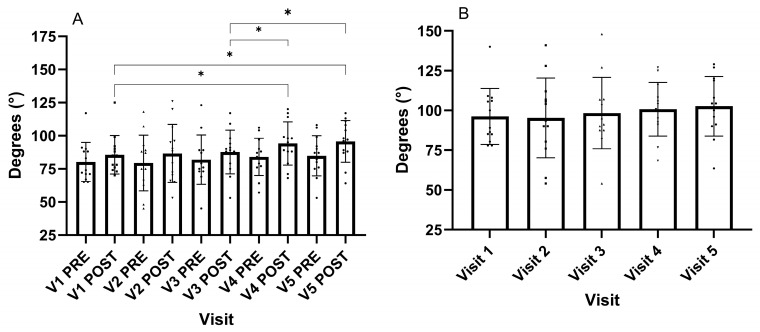
(**A**) ROM achieved to the point of discomfort pre- and post-static stretching at 120% of the point of discomfort. (**B**) Static stretching degrees for the 120% point of discomfort for the five visits. * Different *p* < 0.05. Data presented as group mean and standard deviation with individual responses.

**Figure 2 muscles-04-00033-f002:**
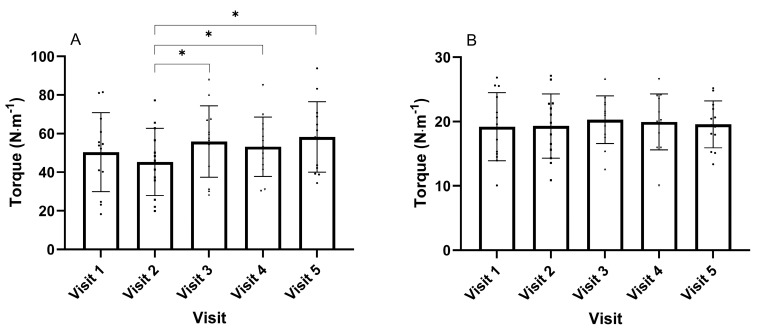
(**A**) Peak torque achieved during the stretches reaching 120% of the point of discomfort; (**B**) gravity-corrected passive stiffness following static stretching. * Different *p* < 0.05. Data presented as group mean and standard deviation with individual responses.

**Figure 3 muscles-04-00033-f003:**
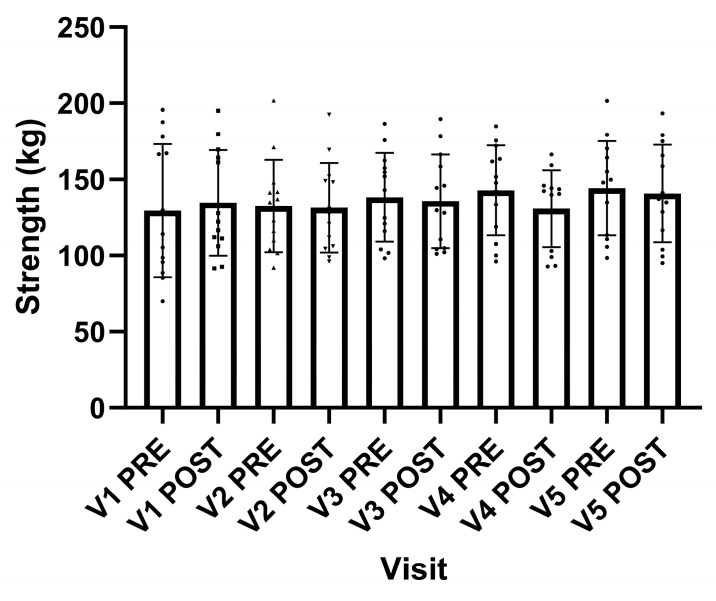
Maximal voluntary isometric contraction of the knee flexors at 50% range of motion pre- and post-30 s static stretching at 120% point of discomfort. Data presented as group mean and standard deviation with individual responses.

**Figure 4 muscles-04-00033-f004:**
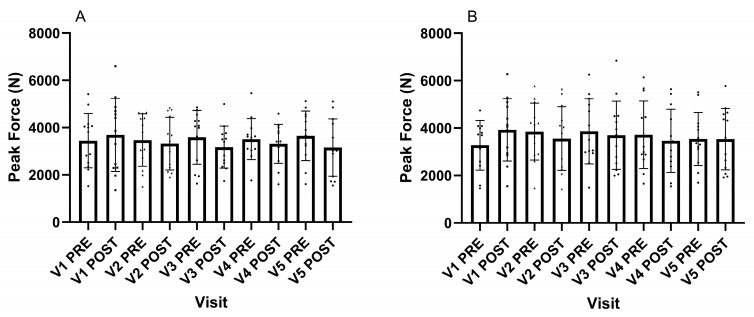
(**A**) Peak force from a single-leg vertical jump off a 20-centimetre box, pre- and post-static stretching at 120% of the point of discomfort. (**B**) Peak force from a single-leg fourth maximal jump immediately following 3 hops, pre- and post-static stretching at 120% of the point of discomfort. Data presented as group mean and standard deviation with individual responses.

**Figure 5 muscles-04-00033-f005:**
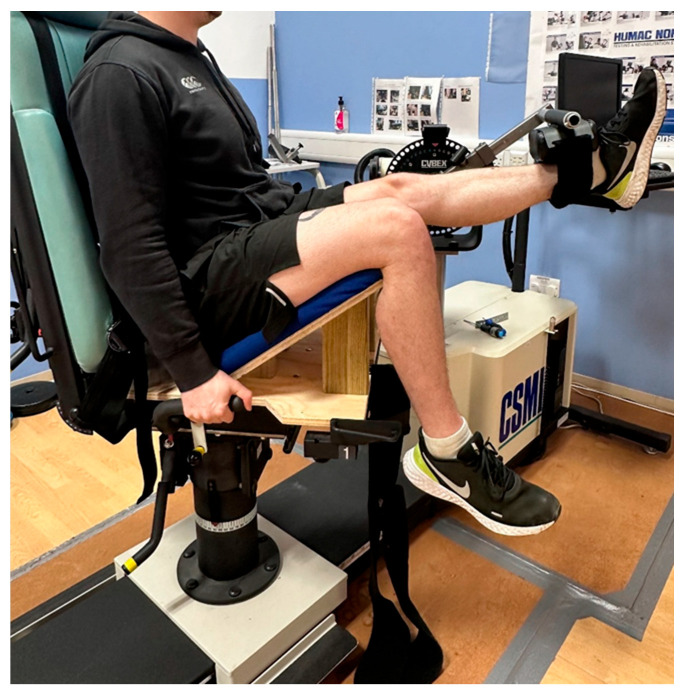
Participant seated on the isokinetic dynamometer performing a static stretch at 120% PoD.

**Table 1 muscles-04-00033-t001:** Reliability as assessed by intraclass correlation and 95% confidence interval (CI) for knee extension ROM, peak MVC and passive stiffness.

	Visit 2–1	Visit 3–2	Visit 4–3	Visit 5–4	Pooled ICC
Knee extension ROM	0.71 (0.37–0.88)	0.90 (0.75–0.96)	0.89 0.72–0.96	0.84 (0.62–0.94)	0.82 (0.68–0.92)
Knee flexion MVC	0.87 (0.68–0.85)	0.69 (0.34–0.87)	0.78 (0.50–0.91)	0.85 (0.63–0.94)	0.81 (0.66–0.91)
Passive stiffness	0.86 (0.67–0.95)	0.85 (0.65–0.94)	0.86 (0.66–0.95)	0.74 (0.43–0.89)	0.81 (0.71–0.93)

## Data Availability

The data presented in this study are available upon request from the corresponding author (MC).

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
