# Peer review of "Reliability of Standardised High-Intensity Static Stretching on the Hamstrings over Multiple Visits"

_muscles, 2025, doi:10.3390/muscles4030033_

Round 1

Reviewer 1 Report

Comments and Suggestions for Authors

This manuscript presents a well-structured repeated-measures study examining the reliability of high-intensity static stretching (120% point of discomfort) on hamstring flexibility, passive stiffness, and strength across five laboratory visits. The topic is relevant, the design is well executed, and the data are clearly presented.

The study contributes novel insights into the consistency of high-intensity SS application, particularly using objective equipment (isokinetic dynamometer). The results will be useful for sport scientists and clinicians.

However, a few aspects require minor clarification:

  1. The increased post-stretch ROM at visits 4 and 5 is presented as a potential training response. Given the low total stretch volume (150 seconds), this should be interpreted cautiously or further substantiated with references.

  2. While VAS scores remained stable, pain tolerance can vary considerably. Please emphasize the limitations of using POD as a target intensity across individuals and studies.

  3. The study only included physically active males. Please explicitly acknowledge in the Discussion that findings may not generalize to females, elite athletes, or clinical populations.

  4. The drop jump and 4th-jump force data were unchanged. Clarify whether this reflects measurement consistency or absence of a stretching effect.

  5. Expressions like “120% of POD” should be explained more clearly for general readers.

Author Response

Comment 1: This manuscript presents a well-structured repeated-measures study examining the reliability of high-intensity static stretching (120% point of discomfort) on hamstring flexibility, passive stiffness, and strength across five laboratory visits. The topic is relevant, the design is well executed, and the data are clearly presented.

The study contributes novel insights into the consistency of high-intensity SS application, particularly using objective equipment (isokinetic dynamometer). The results will be useful for sport scientists and clinicians.

Response 1: Thank you to the reviewer for this comment. The authors appreciate your time in reviewing the manuscript and are grateful for your comments. We highlight below how these have been addressed.

Comment 2: The increased post-stretch ROM at visits 4 and 5 is presented as a potential training response. Given the low total stretch volume (150 seconds), this should be interpreted cautiously or further substantiated with references.

Response 2: Thank you to the reviewer for this comment and we agree that it is difficult to interpret if there was any training response occurring within the present study. We have expanded our point within the manuscript and give further information from the cited study by Thomas et al., (2018).

Comment 3: While VAS scores remained stable, pain tolerance can vary considerably. Please emphasize the limitations of using POD as a target intensity across individuals and studies.

Response 3: Thank you to the reviewer for this comment. The authors agree this is a limitation of the present study, but it is a consistent method used within literature. To address the reviewer’s point, the authors have added a paragraph within the limitations section discussing the point of discomfort limitation.

Comment 4: The study only included physically active males. Please explicitly acknowledge in the Discussion that findings may not generalize to females, elite athletes, or clinical populations.

Response 4: We thank the review for this comment. The authors already had some discussion of the limitation of only have males in the present study. The authors have added that we do not know the effects in clinical populations or females. The authors have also added the recent review article from Elliot-Sale et al., (2021) to support the recommendation for including females in future stretching research.

Comment 5: The drop jump and 4th-jump force data were unchanged. Clarify whether this reflects measurement consistency or absence of a stretching effect.

Response 5: The authors thank the reviewer for this comment and the authors have added these findings to the discussion section. The authors do not know if there is measurement consistency or an absence of a stretching effect, and this has also been added to the discussion. Furthermore, the authors have also added observations from Simic et al., (2013) that static stretching can lead to decreases in power.

Comment 6: Expressions like “120% of POD” should be explained more clearly for general readers.

Response 6: We thank the reviewer for the opportunity to increase clarity in our manuscript. We have added within the introduction section a brief explanation of the terms and how it is used throughout the manuscript.

Reviewer 2 Report

Comments and Suggestions for Authors

I would like to thank the authors for their well-conceived and clearly written manuscript entitled “Reliability of standardised high intensity static stretching on the hamstrings over multiple visits.” The study addresses a relevant and underexplored topic in the field of sports science, namely the reproducibility of high-intensity static stretching across multiple testing sessions. The research question is well formulated, the methodology is sound, and the statistical analyses are generally appropriate. The findings offer meaningful insights for both scientific and applied contexts.

However, several clarifications and minor revisions are necessary to improve the transparency, structure, and practical relevance of the manuscript. These include providing greater detail on participant selection criteria, statistical procedures, the rationale behind methodological choices, and the inclusion of more concrete practical recommendations in the abstract and conclusion.

In light of the above, I recommend minor revisions before the manuscript can be considered for publication.

Line 12:

“Thirteen physically active males”

Please clarify what is meant by “physically active males.” It would be beneficial to define the inclusion criteria in terms of activity level, exercise modality, frequency, and/or training history to enhance the characterization of the study population

Please consider adding practical recommendations at the end of the abstract, highlighting how the study findings may inform training, rehabilitation, or performance monitoring strategies.

Lines 31–34

Please cite all referenced organizations collectively at the end of the sentence to improve clarity and citation style consistency.

Lines 59-60

Please provide a brief rationale for selecting five repeated testing sessions. Was this number informed by existing literature, reliability testing guidelines, or logistical/practical constraints?

General comment on manuscript structure:
The current organization of the manuscript—Introduction, Results, Discussion, Limitations, Materials and Methods, Conclusions—deviates from the conventional scientific format. For clarity and consistency with standard reporting practices, I recommend restructuring the manuscript as follows: Introduction, Materials and Methods, Results, Discussion, Limitations, Conclusions.

Lines 184–188:

Please specify the inclusion criteria used in this study. Additionally, clarify who determined whether participants presented with hypermobility, neurological health conditions, or other exclusionary factors was this based on clinical screening, self-report, or another form of assessment?

191-192

Please justify the choice of a 72-hour minimum interval between testing sessions. Was this duration selected based on previous research, considerations related to recovery and neuromuscular adaptation, or other methodological guidelines?

Line 206-207

To improve the visual clarity and overall engagement of the manuscript, it is recommended to include a schematic figure or illustration showing the experimental setup and participant positioning on the isokinetic dynamometer during the stretching protocol.

Line 217
Please clarify the identity and professional background of the investigator who administered the passive stretching (e.g., trained researcher, physiotherapist). Providing this information would strengthen the methodological transparency and credibility of the procedures employed.

Line 243

Please indicate the specific type of post hoc test applied in the pairwise comparisons (e.g., Bonferroni, Tukey’s HSD, Sidak). This detail is essential for assessing the control of Type I error and the appropriateness of the statistical approach.

It is recommended that the Conclusions section be expanded to include practical implications derived from the findings. For instance, how might the demonstrated reliability of high-intensity static stretching across sessions be applied in athletic training, rehabilitation, or clinical assessment protocols? Providing such recommendations would enhance the translational value of the study.

Author Response

Comment 1: I would like to thank the authors for their well-conceived and clearly written manuscript entitled “Reliability of standardised high intensity static stretching on the hamstrings over multiple visits.” The study addresses a relevant and underexplored topic in the field of sports science, namely the reproducibility of high-intensity static stretching across multiple testing sessions. The research question is well formulated, the methodology is sound, and the statistical analyses are generally appropriate. The findings offer meaningful insights for both scientific and applied contexts.

However, several clarifications and minor revisions are necessary to improve the transparency, structure, and practical relevance of the manuscript. These include providing greater detail on participant selection criteria, statistical procedures, the rationale behind methodological choices, and the inclusion of more concrete practical recommendations in the abstract and conclusion.

In light of the above, I recommend minor revisions before the manuscript can be considered for publication.

Response 1: We thank the reviewer for their time to review our manuscript, their comments and the opportunity to revise the manuscript. We respond to each of the reviewers’ comments below.

Comment 2: Line 12: “Thirteen physically active males” Please clarify what is meant by “physically active males.” It would be beneficial to define the inclusion criteria in terms of activity level, exercise modality, frequency, and/or training history to enhance the characterization of the study population.

Response 2: we thank the reviewer for this comment. The reviewer highlights this in the abstract, however, to better address the comment, the authors have given more detail about the participant population in the methods section. This keeps the word count within the abstract low but gives more detail in the methods section. The authors have added “Participants were all physically active and met or exceeded 150-minutes of physical ac-tivity per week, as determined by healthy history screening. While some participants undertook sport specific training, all the participants were categorized as performance tier 1 (recreationally active) according to the guidelines from McKay et al., [31]”.

 Comment 3: Please consider adding practical recommendations at the end of the abstract, highlighting how the study findings may inform training, rehabilitation, or performance monitoring strategies.

 Response 3: We thank the reviewer for this comment and the authors have added a consideration at the end of the abstract that the findings are not clear when viewed in the context of an applied environment.

Comment 4: Lines 31–34 Please cite all referenced organizations collectively at the end of the sentence to improve clarity and citation style consistency.

Response 4: We thank the reviewer for this comment and the authors have moved the references to the end of the sentence to improve clarity.

Comment 5: Lines 59-60 Please provide a brief rationale for selecting five repeated testing sessions. Was this number informed by existing literature, reliability testing guidelines, or logistical/practical constraints?

Response 5: We thank the reviewer for this comment. The authors have added a brief rationale for the repeated testing sessions within the introduction. These have also been supported by literature to justify using multiple visits for the present study.

Comment 6: General comment on manuscript structure: The current organization of the manuscript—Introduction, Results, Discussion, Limitations, Materials and Methods, Conclusions—deviates from the conventional scientific format. For clarity and consistency with standard reporting practices, I recommend restructuring the manuscript as follows: Introduction, Materials and Methods, Results, Discussion, Limitations, Conclusions.

 Response 6: we thank the reviewer for this comment; however, the authors have followed the author guidelines from the journal on style and layout. If the editorial office of the journal would prefer us to follow the traditional layout, we will be happy to follow the traditional style.

Comment 7:  Lines 184–188: Please specify the inclusion criteria used in this study. Additionally, clarify who determined whether participants presented with hypermobility, neurological health conditions, or other exclusionary factors was this based on clinical screening, self-report, or another form of assessment?

Response 7: the authors thank the reviewer for this comment and the authors have added that participants completed a health history questionnaire and this was used to screen against the exclusion criteria.

Comment 8: 191-192 Please justify the choice of a 72-hour minimum interval between testing sessions. Was this duration selected based on previous research, considerations related to recovery and neuromuscular adaptation, or other methodological guidelines?

Response 8: we thank the reviewer for this comment and we have added some justification for the use of the timings of the visits for this study. We wanted each participant to complete all their laboratory visits within a two-week period so this has been added to the experimental design section.

Comment 9: Line 206-207 To improve the visual clarity and overall engagement of the manuscript, it is recommended to include a schematic figure or illustration showing the experimental setup and participant positioning on the isokinetic dynamometer during the stretching protocol.

Response 9: the authors thank the reviewer for this opportunity to improve the clarity of the manuscript and have added a figure demonstrating the experimental setup and participant positioning on the isokinetic dynamometer.

Comment 10: Line 217 Please clarify the identity and professional background of the investigator who administered the passive stretching (e.g., trained researcher, physiotherapist). Providing this information would strengthen the methodological transparency and credibility of the procedures employed.

Response 10: the authors thank the reviewer for this comment and we have added details about the primary investigator who undertook all the experimental protocols.

Comment 11:  Line 243 Please indicate the specific type of post hoc test applied in the pairwise comparisons (e.g., Bonferroni, Tukey’s HSD, Sidak). This detail is essential for assessing the control of Type I error and the appropriateness of the statistical approach.

Response 11: we thank the review for this comment and we have confirmed that post hoc testing was performed with Bonferroni correction. We have also added detail on the power analysis undertaken for this experiment.

Comment 12: It is recommended that the Conclusions section be expanded to include practical implications derived from the findings. For instance, how might the demonstrated reliability of high-intensity static stretching across sessions be applied in athletic training, rehabilitation, or clinical assessment protocols? Providing such recommendations would enhance the translational value of the study.

Response 12: We thank the reviewer for this opportunity to expand upon the practical implications from our experiment. We have added some of these to the conclusions section.
